# Metal-Organic Framework Assembled on Oriented Nanofiber Arrays for Field-Effect Transistor and Gas Sensor-Based Applications

**DOI:** 10.3390/molecules27072131

**Published:** 2022-03-25

**Authors:** Jinlong Mu, Xing Zhong, Wei Dai, Xin Pei, Jian Sun, Junyuan Zhang, Wenjun Luo, Wei Zhou

**Affiliations:** Faculty of Materials Science and Chemistry, China University of Geosciences, 388 Lumo Road, Wuhan 430074, China; mujl1122@foxmail.com (J.M.); zx13511175236@gmail.com (X.Z.); dw15387090924@163.com (W.D.); pxin0111@gmail.com (X.P.); sunjian@cug.edu.cn (J.S.); mufosen@126.com (J.Z.)

**Keywords:** metal–organic frameworks, liquid epitaxial growth, oriented nanofiber arrays, field-effect transistor, gas sensor

## Abstract

Metal–organic framework (MOF) films are essential for numerous sensor and device applications. However, metal-organic framework materials have poor machinability due to their predominant powder-like nature, and their presence as the active layer in a device can seriously affect the performance and utility of the device. Herein, active layers of field-effect transistor (FETs) devices and chemiresistor gas sensors with high performance were constructed by loading Cu_3_(HITP)_2_ (HITP = 2,3,6,7,10,11-hexaiminotriphenylene) in situ-axial anchoring on oriented nanofiber arrays prepared via electrospinning. The strong interaction between polar groups on the polymer chains and metal ions promotes the nucleation of Cu_3_(HITP)_2_, steric hindrance makes particles of Cu_3_(HITP)_2_ with uniform size, morphology, and good crystallinity during nucleation by liquid phase epitaxial growth (LPE). Influences of differently-oriented Cu_3_(HITP)_2_ NFAs-based FETs on the electrical properties were studied, optimally oriented Cu_3_(HITP)_2_ NFAs-based FETs showed good mobility of 5.09 cm^2^/V·s and on/off ratio of 9.6 × 10^3^. Moreover, excellent gas sensing response characteristics were exhibited in sensing volatile organic compounds (VOCs). Chemiresistor gas sensors with high response value, faster response and recovery are widely suited for VOCs. It brings new inspirations for the design and utilization of electrically conductive MOFs as an active layer for FETs and sensor units for chemiresistor gas sensors.

## 1. Introduction

The development and research of two-dimensional (2D) layered materials have been a topic of great interest to researchers since graphene was first reported [1]. Of special interest is the high aspect ratio of 2D metal–organic framework (MOF) porous materials, which consist of a stable network structure based on organic connectors between metal nodes. The inherent band structure and electronic properties of 2D MOFs can be systematically adjusted by the coordination environment, organic ligands, and reaction conditions of molecular building blocks in the assembly process. Compared with traditional inorganic materials, the organic groups and metal diversity give 2D MOFs interesting structure and electronic properties, enabling them to have structural flexibility and functional diversity [2].

The plane of 2D MOFs is usually stacked face-to-face. The functional groups in adjacent layers have strong π-π interactions, which facilitates charge migration, but MOFs are rarely applied to FETs because most are electrical insulators [3]. In recent years, 2D-layered MOF-coordinated polymers with strong metal–ligand orbital hybridization are characterized by a very distinctive high conductivity, such as MOFs of the M_3_(hexaiminotriphenylene)_2_ (M = Ni, Cu, Co) class. M_3_(HITP)_2_ (M = Cu, Ni) is a charge-neutral material composed of bivalent transition metal cations (Ni^2+^, Cu^2+^, Co^2+^) and HITP ligands, of which Ni_3_(HITP)_2_ shows the highest conductivity of microporous MOFs reported to date. Cu_3_(HITP)_2_ (HITP = 2,3,6,7,10,11- hexaiminotriphenylene) also shows high conductivity and can be found in transistors [4,5], gas sensors [6,7], supercapacitors [8], lithium–sulfur batteries [7], and heterogeneous catalysts [9,10]. The Cu-HITP 2D shows high electrical conductivity attributed to charge delocalization in the 2D plane or extended π conjugations along with the stacked columns [11,12]. The microporous honeycomb structure is formed along the c-axis according to the AB plane of slight slip. Compared with traditional MOFs, 2D-layered MOFs with planar coordination metal ions and π-π-conjugated ligands have very high conductivity. However, the short-range interactions and large particle size of most MOFs inhibit the rapid transport of electrons and holes, the performance stability of the device is adversely affected by the unstable connection with the electrode, which necessitates that newer devices integrate materials that can meet higher requirements for both stability and application performance. The study of high-quality continuous MOF films is a promising avenue for further improvements in the performance of these devices. In addition, the organic semiconductor is in the form of “molecular solid”, which makes the semiconductor layer more susceptible to the influence of the insulating layer, resulting in significant changes in carriers and mobility. It is generally believed that the charge accumulation area is distributed at the two-dimensional interface between the semiconductor and the insulating layer. So the greater the dielectric constant is, the larger the specific capacitance that can be obtained under the same conditions, which can enable the device to open at a lower threshold voltage. Consequently, the ideal insulating layer material plays a decisive role in improving the performance of the device. In recent years, self-assembled monolayer (SAM) and self-assembled multilayer (SAMT) have attracted the attention of researchers due to their excellent dielectric properties, controllable thickness, and molecular diversity.

In 2007, Woll and co-workers were the first to apply layer-by-layer deposition to the growth of highly oriented and crystalline MOF thin films on self-assembled monolayer (SAM)-modified Au substrates [13]. Since then, a series of reports have also demonstrated how active layer growth can be regulated via the dielectric layer by using a bottom-up strategy to prepare two-dimensional metal complexes. Recently, significant progress has been made in researching layer-by-layer (LBL) multilayer assemblies for interface engineering of organic optoelectronic devices that show great potential in practical application [14]. The liquid phase epitaxy (LPE) layer-by-layer approach efficiently controls the growth orientation, thickness, and homogeneity of MOF thin films produced through LBL [3,15,16], making it possible to use them in devices.

This category of MOF films (called surface-mounted MOFs, or SURMOFs) shows promise in sensor and device applications. Layer-by-layer assembly techniques are used to grow SURMOFs on self-assembled monolayer (SAM)-functionalized substrates. The process involves repeatedly and alternately immersing a surface-functionalized base in solutions of inorganic metal ions or metal cluster connectors and solutions of multi-dentate organic ligands. Between these two steps, the substrate is cleaned with pure solvents to remove uncoordinated metal or organic joints [17]. Similarly, the dipping method [13], spraying method [18], and pumping method [19] were also carried out as bottom-up LBL growth methods. A variety of uniform and stable MOF films were prepared, such as heat-assisted solvent-evaporation in situ growth on the porous substrate as a template [20]. An ordered MOF-on-MOF was obtained by combining organic ligands in situ with the metal point of another MOF as the substrate [21].

In addition to device stability, the response rate of the device is also the main factor in evaluating device performance. There are inevitably many grain boundaries and defects in the thin film on the substrate surface, which seriously affects device performance. Compared with thin films, micro/nanowires have unique advantages. Microscopically, the π-π stacking direction parallel to the substrate, along the long axis of the micro/nanowires, is beneficial to carrier transport [22]. Macroscopically, shortening the carrier transport distance between electrodes can improve the response rate of the device.

The directional transmission of a semiconductor micro/nano line carrier determines if it can be used as an organic micro/nano construction unit. Integrated circuits have a significant advantage in the miniaturization and integration of devices. The micro/nano line possesses considerable length relative to its diameter, and because of this, the growth of micro/nanowires occurs along the direction of intermolecular π-π [23]. In this way, the conductivity of micro/nanowire shows anisotropy. The path along the micro/nanowires is conducive to the migration of carriers and the realization of directional charge transmission. However, preparing a highly oriented transport layer perpendicular to the electrode presents significant challenges.

Electrospinning is a simple and inexpensive method for the large-scale preparation of nanofibers of different sizes. Through simple modification, electrospinning can achieve highly ordered nanofiber arrays (NFAs) on various substrates as well [24]. With functional fibers as a template, the active layers are arranged perpendicular to the electrode to shorten carrier migration distance and accelerate device response. This enables the exciting prospect of the preparation of highly ordered NFAs with multiple active sites by electrospinning on the device substrate to make two-dimensional MOFs grow in situ on the fiber surface.

In this work, a brand-new strategy for in situ growth of highly oriented MOF NFAs is first designed and proposed. The oligomer used, consisting of 2,6-pyridinedicarboxamide ligands that coordinate to Cu(II) centers, was employed as a functionalized substrate for Cu_3_(HITP)_2_ in situ growth by liquid phase epitaxial growth (LPE). The performance of gas sensors and FETs with different orientations of Cu_3_(HITP)_2_ NFAs as active layers were discussed and anticipated.

## 2. Experimental Section

Figure 1 shows the Cu_3_(HITP)_2_ in situ growth on functionalized cross-linked coordination compound fiber arrays by LBL. This preparation can be briefly summarized into three steps: synthesis of the fiber precursor, preparation of PDMS-PCDA/Cu NFAs, and in situ growth of the NFAs. The process specifics are as follows:

### 2.1. Synthesis of the Oligomer Ligands PDMS-PCDA/Cu

Triethylamine (Et_3_N) was added to a certain amount of H_2_N-PDMS-NH_2_ (Mn = 6000)-dichloromethane (CH_2_Cl_2_) solution at 0 °C under argon atmosphere. After full agitation, the solution of 2,6-pyridinedicarboxamide (PCDA) in CH_2_Cl_2_ was slowly added. The final product, PDMS-PCDA/Cu, was prepared by adding a certain amount of copper sulfate–methanol solution into the PDMS-PCDA CH_2_Cl_2_ solution after continuous stirring and concentration.

### 2.2. Preparation of the Oriented PDMS-PCDA/Cu NFAs

The PDMS-PCDA/Cu was dissolved in a certain amount of polymethyl methacrylate (PMMA)-tetrahydrofuran (THF) solution and thoroughly stirred. Next, a certain amount of dimethylformamide (DMF) solution was added, and the resulting solution was used for electrostatic spinning. The (n++) silicon wafer substrate was fixed on the roller’s surface, the roller’s rotation speed was adjusted (300 r/min, 400 r/min, or 500 r/min), and the PDMS-PCDA/Cu NFAs were obtained on the (n++) silicon wafer substrate. The PDMS-PCDA/Cu solution formed nanofibers under the action of an electric field force, and the PDMS-PCDA/Cu NFAs were collected on the roller and substrate.

### 2.3. Cu_3_(HITP)_2_ In Situ Growth on PDMS-PCDA/Cu NFAs

The prepared PDMS-PCDA/Cu NFAs substrate was placed first in an aqueous ammonia solution, then in a HITP organic ligand solution to complete the growth of the first Cu_3_(HITP)_2_ MOF layer. The substrate was repeatedly exposed to copper salts in ammonia solution followed by organic aqueous solutions (the LBL method) to achieve rapid production of Cu_3_(HITP)_2_ NFAs (20 growing cycles). See the Appendix A for further details (Appendix A. Materials and Methods).

## 3. Results and Discussion

To further probe the chemical structure of cross-linked coordination oligomer PDMS-PCDA components, Fourier transform infrared spectroscopy (FT-IR) was applied. As shown in Figure 2a, the peaks at 3423 cm^−1^ were mainly ascribed to amino-capped PDMS in the Si-NH_2_ structure [25], and the characteristic peaks at 2963 cm^−1^ and 2904 cm^−1^ were derived from -CH_3_ stretching vibrations. The vibration of the Si-O-Si skeletons had a peak between 1207 cm^−1^ and 925 cm^−1^, and the vibration of Cl-C=O bonds from PCDA showed a peak at 1750 cm^−1^. As can be seen from the PDMS-PCDA infrared spectrum, the amino (Si-NH_2_) characteristic peak at 3423 cm^−1^ disappeared. The new peak at 1536 cm^−1^ is from stretching vibrations of the amide II band, which indicates the conversion of acyl chlorides to amides during condensation. As a result of the bond conversion, the bond energy has also been blue-shifted from 1750 cm^−1^ to 1662 cm^−1^ or 1683 cm^−1^. These results indicate the successful synthesis of PDMS-PCDA cross-linked coordination oligomers, which is consistent with additional results from nuclear magnetic resonance (^13^C NMR spectra and ^1^H NMR spectra) (Appendix A). The strong metal–ligand binding site pyridine N and the weak metal–ligand binding site C=O in PDMS-PCDA is adjacent, enabling the formation of stable binding sites with copper ions in the polymer. These sites are firm and evenly dispersed, providing necessary precursor conditions for in situ growth and ordered arrangement of MOFs.

The elemental compositions and valence states of PDMS-PCDA/Cu NFAs were investigated by X-ray photoelectron spectroscopy (XPS) survey spectrum, as shown in Figure 2b. The high-resolution XPS spectrum of C1s can be divided into three peaks—284.8 eV, 285.7 eV, and 286.48 eV, which can be assigned to C=C, C-N, and C=O, respectively. In addition, the high-resolution XPS spectrum of N1s consists of one peak, including NH and N=C. The XPS spectrum of the Cu(2p) region suggests that copper forms coordination bonds with pyridine N and C=O in the Cu(I) valence state, resulting in two splitting peaks (952.3 eV and 932.41 eV). The optical photo in Figure 2c shows PDMS-PCDA/Cu fiber arrays on the 1 cm × 1 cm (n^++^) Si substrate after electrostatic spinning. The morphology and elemental mappings of C, N, and Cu in the PDMS-PCDA/Cu fiber are shown in Figure 2d. It can be seen that the fiber morphology is uniform in thickness, with a good aspect ratio, and that the surface of the fiber is smooth. All of these elements distribute uniformly along with the fiber. The mapping results indicate that cross-linked coordination oligomers and copper ions achieved good coordination and that the elements are evenly and reliably dispersed in the fibers.

Morphologies and chemical compositions of PDMS-PCDA/Cu_3_(HITP)_2_ NFAs were characterized by optical photography, scanning electron microscopy (SEM), energy dispersion spectrum (EDS), IR spectroscopy, and X-ray photoelectron spectroscopy (XPS). Internal structures were revealed by transmission electron microscopy (TEM) and X-ray diffraction (XRD). PDMS-PCDA/Cu NFAs was used as the template. Each *o*-phenylenediamine link was expected to be oxidized to a radical anion form by an aqueous ammonia solution, resulting in a charge-neutral complex with the Cu^2+^ centers on the fibers. Therefore, an oxidized HITP solution and Cu^2+^ completed the initial frame of MOF in situ on PDMS-PCDA/Cu fibers. Next, Cu_3_(HITP)_2_ was grown layer-by-layer on the surface of fibers after repeated treatments with a copper sulfate–ammonia aqueous solution and a HITP aqueous solution. A significant color change can be seen from the optical photo in Figure 3c, where the original milky fiber turns black (the process of change from PDMS-PCDA/Cu fiber membrane to Cu_3_(HITP)_2_ fiber membrane is shown in Appendix A). Figure 3d shows the SEM images and EDS of Cu_3_(HITP)_2_ grown on the fibers; these nanosheets are closely compacted, making the in situ grown nanosheets very dense with fibers. According to EDS, the distributions of C, N, and Cu are uniform across the fiber surfaces. Compared with the previously discussed PDMS-PCDA/Cu fiber, PDMS-PCDA/Cu_3_(HITP)_2_ fibers show an increased density of the three elements with the same number of sweeps.

The FT-IR of both PDMS-PCDA/Cu and PDMS-PCDA/Cu_3_(HITP)_2_ are shown in Figure 3a. Compared with the FT-IR curve of PDMS-PCDA/Cu fiber, a new characteristic peak appears for PDMS-PCDA/Cu_3_(HITP)_2_ at 1144 cm^−1^, which is attributable to the *o*-phenylenediamine bond (CH_2_-NH) from Cu_3_(HITP)_2_ NFAs. The peak at 500 cm^−1^ disappeared, which may be caused by the oxidation of copper ions to form a new coordination bond. This illustrates that Cu_3_(HITP)_2_ has grown on the surface of the PDMS-PCDA NFAs. Similarly, Figure 3b, the XPS spectrum obtained for Cu_3_(HITP)_2_ NFAs, indicates the presence of C1s, including C=C, C-N, and C=O. In addition, the high-resolution XPS spectrum of N1s still shows a single type of N, confirming that additional NH_4_^+^ cations are also absent [26] and that there are no residual unreacted precursors and solvents from the LBL growth process. The high resolution of Cu suggests a mixture of Cu(I) and Cu(II) centers of two splitting peaks, Cu2p3/2 (955 eV, 952.79 eV) and Cu 2p1/2 (933.4 eV, 935.3 eV), which suggests that the redox-active HITP ligands compensate for the variation from Cu^2+^.

Powder X-ray diffraction (PXRD) analysis revealed that Cu_3_(HITP)_2_ adopts a hexagonal 2D structure with slipped-parallel stacking of the 2D sheets. The structural model of the simulation is shown in Figure 4a; unit parameters of the simulated structure are: a = 22.02 Å, b = 22.02 Å, c = 6.6 Å, α = 90°, β = 90°, γ = 120°. The XRD spectra of both Cu_3_(HITP)_2_ powder and Cu_3_(HITP)_2_ NFAs were consistent with the spectrum of simulated Cu_3_(HITP)_2_, with peaks located at 2θ = 4.63°, 9.26°, and 27.4°, corresponding to the (100), (200), and (001) planes, respectively (Figure 4b). This shows that Cu_3_(HITP)_2_ powder and Cu_3_(HITP)_2_ NFAs have similar crystal structures. The width of the peak at 2θ = 27.4° corresponds to both the amorphous polymer and the reflection of [001], a sign that the long program along the C direction is poor compared to the AB plane. This is typical of layered 2D materials and further corresponds to the image in the SEM, in which the MOF nanosheet grows along the AB plane.

The interface between PDMS-PCDA/Cu fibers and Cu_3_(HITP)_2_ NFAs was observed by high-resolution TEM (HR-TEM). As shown in Figure 4c, there is an apparent black interface between the fiber interior and the edge, with a width of about 50 nm. At this resolution, it can be seen that the fiber junction is amorphous in an inward direction and that the crystal diffraction pattern has an outward direction. The TEM image demonstrates that MOF is grown in situ at the outer edges of the PDMS-PCDA/Cu fiber (black). Additionally, the high-resolution (HR-TEM) image in Figure 4d was recorded from Figure 4c and is marked by a red circle. The fast Fourier transformation (FFT) image recorded the area that is marked by small red boxes in Figure 4d, it revealed a hexagonal crystal structure. For the in situ-grown Cu_3_(HITP)_2_, the distance between the crystal planes is 1.9 nm, which corresponds to the (100) crystal planes of Cu_3_(HITP)_2_. This also signals that the growth of Cu_3_(HITP)_2_ on the fiber surface has good crystallinity and that the AB direction of π-π stacking grows along the long axis of the micro/nanowires, which is conducive to carrier transport. To explore potential applications of Cu_3_(HITP)_2_ NFAs in electronics, FETs, and gas sensors were fabricated with Cu_3_(HITP)_2_ NFAs as the active layer and sensor unit.

### 3.1. Field-Effect Transistor Measurements

An interdigital electrode was used as the mask to deposit Au and form electrode arrays. The schematic diagram of top-contact Cu_3_(HITP)_2_ NFAs-based FET devices fabricated on a highly phosphorus-doped silicon (n^++^) Si substrate is shown in Figure 5a (SEM images of Cu_3_(HITP)_2_ NFAs-based FET are shown in Appendix A).

The transfer characteristics of variously oriented Cu_3_(HITP)_2_ NFAs-based FETs (Figure 5b–d) all show an enhancement in drain current (I_ds_) while applying negative gate voltage (V_g_). Characteristics of the curves are evidence of typical p-type channel material present between source and drain; V_g_ measures the flow of charge carriers from the source to the drain. Device performance data of differently-oriented Cu_3_(HITP)_2_ NFAs-based FET (20-cycles for Cu_3_(HITP)_2_ growth) are listed in Table 1.

Transfer curves show that the current switch ratio (I_on_/I_off_) of FETs obtained at 300 r/min, 400 r/min, and 500 r/min (rotational speed of drum during electrospinning) increased gradually up to 4.8 × 10^3^, 7.9 × 10^3^, and 9.6 × 10^3^, respectively. The threshold voltage of these FETs was observed to drop after initial cycling and stabilize as low as 3.4 V, 3.05 V, and 2.9 V, respectively. The high ON/OFF current ratio and the low threshold voltage indicates that Cu_3_(HITP)_2_ NFAs-based FETs exhibit exceptional ON/OFF switching performances. The carrier mobility is also an important parameter in assessing the electronic properties of FETs [27]. Transfer curves of three Cu_3_(HITP)_2_ NFAs-based FETs with different fiber orientations demonstrate that the carrier concentration in the channel increases when decreasing the gate voltage from −10 V to 10 V, revealing hole mobilities of 4.23 cm^2^/V·s, 4.41 cm^2^/V·s, and 5.09 cm^2^/V·s. As the placement of the Cu_3_(HITP)_2_ NFAs becomes more perpendicular to the electrode, the carrier transfer path shortens, resulting in a better response of the FET. In addition, the V-I curve of Cu_3_(HITP)_2_ NFAs on the Si (n^++^) substrate was measured by Hall effect measurement (Appendix A) and showed that the direction of Cu_3_(HITP)_2_ NFAs had a good linear V-I curve, but that it was nonlinear in the direction perpendicular to the Cu_3_(HITP)_2_ NFAs. It shows that there is good ohmic contact in the orientation of the Cu_3_(HITP)_2_ NFAs. These demonstrate that the electrical properties of Cu_3_(HITP)_2_ NFAs-based FETs can be further improved by optimizing their orientation. Table 2 summarizes the FET devices’ performance based on the reported conductive MOFs to date [28].

Water vapor not only induces the decomposition of MOF functional layers but also impedes the selectivity of MOF-based sensors. Figure 5e shows the contact angle of water on the silicon and Cu_3_(HITP)_2_ NFAs substrates. The angle increases from 83.19° to 138.88°, respectively, demonstrating the good hydrophobic properties of the Cu_3_(HITP)_2_ NFAs. This allows it to be used in devices that have a broader range of operating conditions.

### 3.2. Chemiresistive Gas Sensors

Cu_3_(HITP)_2_ NFAs were used as gas sensing units to explore their characteristics in sensing volatile organic compounds (VOCs). Figure 6a contains the response–recovery (defined as ΔR/R_0_, the relative change in the resistance ΔR upon gas contact compared to the resistance in high-purity dry air R_0_) curves of the Cu_3_(HITP)_2_ NFAs-based sensor to different concentrations of ethanol at room temperature. When ethanol was introduced into the test tube, the sensor’s resistance rose rapidly and gradually leveled off. When the ethanol in the test tube was replaced by high-purity dry air, the sensor’s resistance quickly fell to the original resistance position.

It is generally believed that the sensing mechanism of MOF gas sensors is based on the influence of gas molecule adsorption/desorption on resistance changes [32,33]. Some researchers also believe that oxygen molecules (O_2_) in the air form oxygen ions (O_2_^−^) by absorbing electrons on the active site of MOFs at a specific temperature. When exposed to a reducing gas, O_2_^−^ on the MOF surface loses electrons back to the MOF films, resulting in increased resistance of the gas sensor [34]. The data presented here show that Cu_3_(HITP)_2_ NFAs at room temperature are typical p-type semiconductors with holes as the primary carrier. The movement of holes as carriers in MOF crystals is limited upon exposure to reducing gases, which supply electrons to fill the holes, reduce the flow of carriers, and increase the resistance state of the gas sensors. In addition, the response and recovery performance of the Cu_3_(HITP)_2_ NFAs-based sensor varied with different ethanol concentrations. The response and recovery time for 20 ppm of ethanol are τ_res_ = 50 s and τ_rec_ = 64 s, respectively (Figure 6b), indicating that the sensor has good response–recovery reversibility.

The sensor did not immediately return to the original resistance after contacting the air, indicating that there is a desorption process for the reducing gas. Figure 6c shows the response (%) vs. concentration of the sensor toward ethanol. The excellent linearity in the 1–50 ppm range is comparable to a typical chemiresistor gas sensor. The theoretical limit of detection (LOD) can be calculated to be about 0.5 ppm from the simulated linear equation by setting the response to be 10%. Figure 6d shows the specific sensitivity of Cu_3_(HITP)_2_ NFAs-based sensors toward 200 ppm of various VOCs at room temperature: acetone, ethanol, xylene, toluene, methylbenzene, methanol, triethylamine, and butanone. The sensor quickly recovered to the original resistance state during the test, and the cycle test was consistent. Loading Cu_3_(HITP)_2_ nanoparticles on fiber arrays led to gradually enhanced gas absorption and diffusion with increasing surface area. This is due to the unique orientation and stacking of Cu_3_(HITP)_2_ grown in situ on the fiber arrays, which enables stronger adsorption of the target gas and ensures a better carrier migration rate and sensor stability. The structure increases the density of the active site, making it easy to react with the target gas and improving the sensor’s sensitivity. Cu_3_(HITP)_2_ NFAs oriented perpendicular to the electrode ensure the efficient transfer of carriers in the sensor layer.

## 4. Conclusions

In summary, we successfully prepared directionally oriented nanofiber arrays of PDMS-PCDA/Cu as dielectric layers and in situ growth templates for MOFs via adjusting the electrospinning parameters. The dielectric layer PDMS-PCDA/Cu NFAs were combined with the organic ligand HITP to form Cu_3_(HITP)_2_ NFAs through in situ growth and the LBL method. This process provides a new and reliable way to prepare highly oriented metal–organic framework (MOF) nanofiber arrays (MNFAs). FET devices using Cu_3_(HITP)_2_ NFAs as a hole transport layer showed high carrier mobility and on/off ratio, low threshold voltages, and outstanding stability. As the higher rpm resulted in the higher orientation of nanofiber arrays, Cu_3_(HITP)_2_ NFAs FET devices showed an increased μ and on/off ratio, which indicates that carriers have higher migration in the MOF nanofiber arrays with higher orientation. It is one of the best performing MOF NFAs-based FETs to have been reported to date. The Cu_3_(HITP)_2_ NFAs-based sensor also shows high sensitivity in gas detection response and recovery. These results demonstrate the viability of stable, efficient MOF electronic devices with coordinating functional dielectric layers, indicating a new method for constructing these devices.

## Figures and Tables

**Figure 1 molecules-27-02131-f001:**
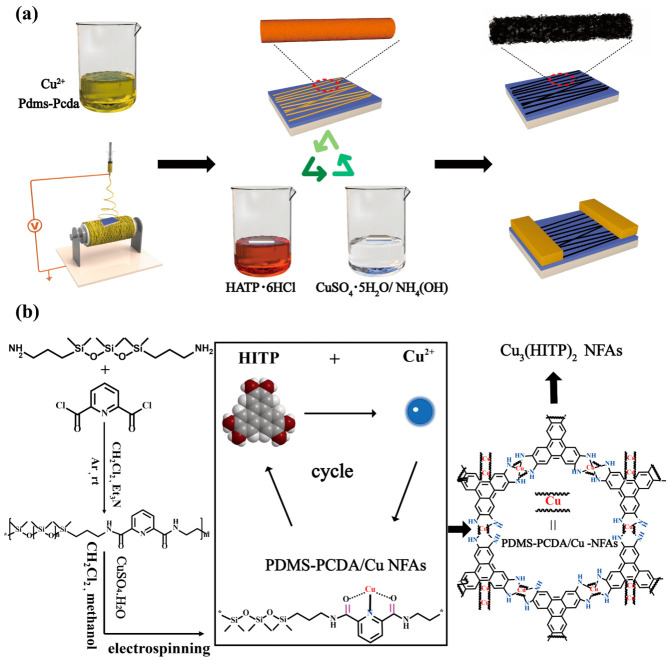
(**a**) Schematic illustration for the layer-by-layer growth procedure of Cu_3_(HITP)_2_ NFAs on functionalized polymer fiber. (**b**) Reaction scheme of Cu_3_(HITP)_2_ NFAs.

**Figure 2 molecules-27-02131-f002:**
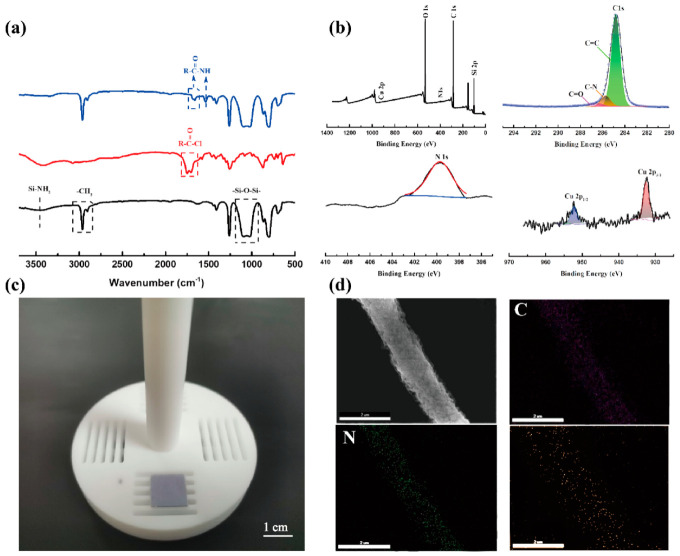
Characterization of PDMS-PCDA/Cu NFAs. (**a**) FT-IR of PDMS (black), PCDA (red), and PDMS-PCDA (blue). (**b**) XPS of PDMS-PCDA/Cu NFAs, focusing on N 1s, C 1s, and Cu 2p. (**c**) Photograph of the PDMS-PCDA/Cu fiber array on the (n^++^) silicon wafer substrate. (**d**) SEM/EDX mapping images of PDMS-PCDA/Cu fiber.

**Figure 3 molecules-27-02131-f003:**
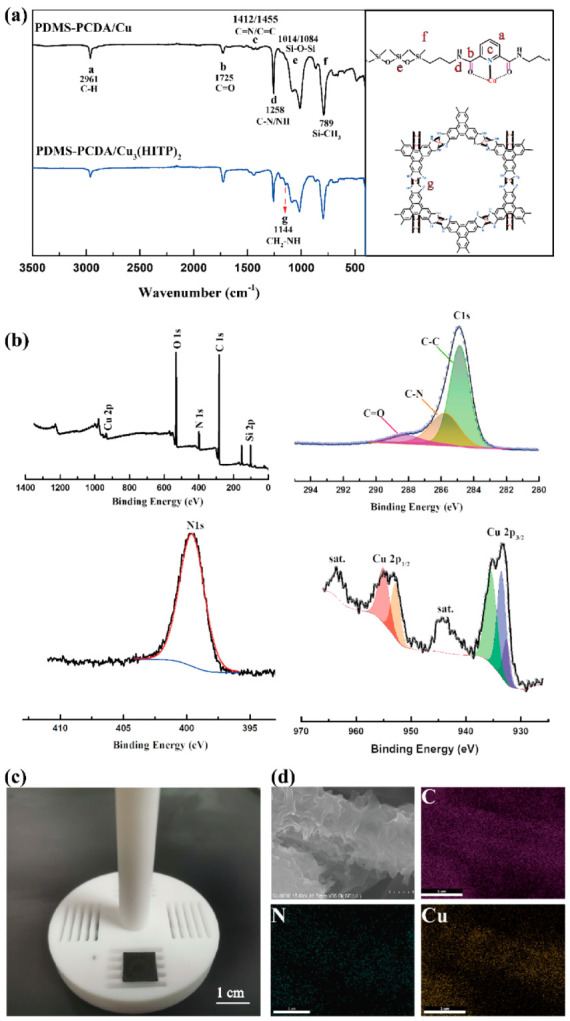
Characterization of PDMS-PCDA/ Cu_3_(HITP)_2_ NFAs. (**a**) FT-IR of PDMS-PCDA/Cu NFAs (black) and PDMS-PCDA/Cu_3_(HITP)_2_ NFAs (blue). (**b**) XPS of PDMS-PCDA/Cu_3_(HITP)_2_ NFAs, focusing on N1s, C1s, and Cu2p. (**c**) Photograph of the PDMS-PCDA/Cu_3_(HITP)_2_ fiber arrays on the (n++) Si substrate. (**d**) SEM/EDX mapping images of PDMS-PCDA/Cu_3_(HITP)_2_ fiber.

**Figure 4 molecules-27-02131-f004:**
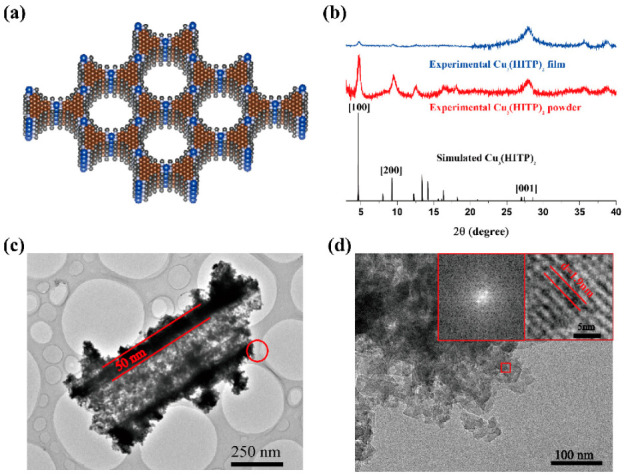
(**a**) Structure of Cu_3_(HITP)_2_ viewed down the c-axis (brown: carbon, black: hydrogen, gray: nitrogen, blue: copper). (**b**) XRD pattern of simulated (black) and experimental Cu_3_(HITP)_2_ powder (red) and PDMS-PCDA/Cu_3_(HITP)_2_ NFAs (blue). (**c**) TEM images of PDMS-PCDA/Cu_3_(HITP)_2_ NFAs. (**d**) HR-TEM (corresponding red circle in (**c**)) and FFT patterns (corresponding red box in (**d**)) of PDMS-PCDA/Cu_3_(HITP)_2_ NFAs.

**Figure 5 molecules-27-02131-f005:**
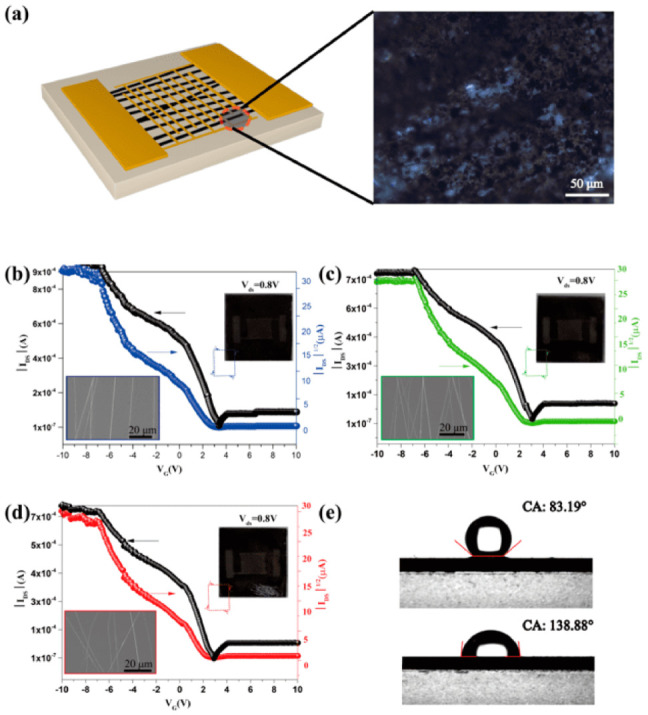
(**a**) Schematic diagram of a top-contact Cu_3_(HITP)_2_ NFAs-based FET device (left), and optical microscopy image of Cu_3_(HITP)_2_ NFAs deposited over a (n^++^) Si substrate. (**b**–**d**) Transfer characteristic for FET devices using Cu_3_(HITP)_2_ NFAs /(n^++^) Si spun at (**b**) 500 r/min, (**c**) 400 r/min, and (**d**) 300 r/min (Illustration: FETs devices diagram in upper right, SEM of Cu_3_(HITP)_2_ NFAs in lower left). (**e**) Contact angle of water on Cu_3_(HITP)_2_ NFAs of (n^++^) Si substrate(above), and (n^++^) Si substrate (below).

**Figure 6 molecules-27-02131-f006:**
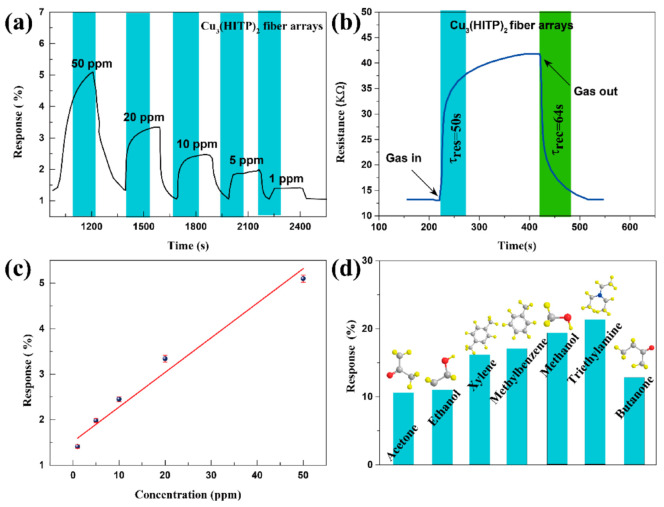
(**a**) Relative responses of a Cu_3_(HITP)_2_ device to 1, 5, 10, 20, and 50 ppm ethanol. (**b**) Response and recovery times defined for Cu_3_(HITP)_2_ fiber arrays sensing 20 ppm of ethanol. (**c**) Device response as a function of ethanol concentration (R^2^ = 0.96). (**d**) Sensing response of Cu_3_(HITP)_2_ fiber array sensors to 200 ppm of acetone, ethanol, xylene, toluene, methylbenzene, methanol, triethylamine, and butanone at room temperature.

**Table 1 molecules-27-02131-t001:** Electrical characteristics of the differently-oriented Cu_3_(HITP)_2_ NFAs-based FET.

The Receiving NFAs of Rotation Speed	μ (cm^2^/(V s))	Vth (V)	I_on_/I_off_
300 r/min	4.23	3.4	4.8 × 10^3^
400 r/min	4.41	3.05	7.9 × 10^3^
500 r/min	5.09	2.9	9.6 × 10^3^

**Table 2 molecules-27-02131-t002:** Summary of conductive MOFs and their FET devices performance reported to date.

Materials [ref]	μ (cm^2^/(V s))	V_th_ (V)	I_on_/I_off_
Cu-BHT [29]	116 (e), 99 (h)	/	10
Ni_3_(HITP)_2_ [4]	48.6 (h)	1.1	2000
Fe_2_(BDP)_3_ [30]	6 × 10^−4^ (h), 2 × 10^−3^ (e)	/	/
K_0.98_Fe_2_(BDP)_3_ [30]	0.84 (e)	/	/
Im@CuBTC [31]	4.0 × 10^−5^ (h)	7	826
Ni_3_(HITP)_2_ [22]	45.4 (h)	−2	2290
Our work	5.09 (h)	2.9	9.6 × 10^3^

e: electron; h: hole.

## Data Availability

The raw/processed data required to reproduce these findings cannot be shared at this time as the data also forms part of an ongoing study.

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
