# Peer review of "Metal-Organic Framework Assembled on Oriented Nanofiber Arrays for Field-Effect Transistor and Gas Sensor-Based Applications"

_molecules, 2022, doi:10.3390/molecules27072131_

Round 1
Reviewer 1 Report
Recommendation: Publish in Molecules after minor revisions.
Comments:
The authors present a robust approach to fabricate a MOF-based nanofiber arrays for the application of gas sensing application, with a FET device fabricated for illustrative purpose. the reviewer found the manuscript is well-structured and informative, therefore is suitable for publish in Molecules after a minor revision.
- In line 53, please provide a reference of “charge delocalization in the 2D plane”.
- In line 135, I believe you mean PDMS-PCDA/Cu instead of PDMS-PCDA, as latter sentence suggested the addition of copper sulfate-methanol solution.
- Please increase contrast and resolution of Cu in panel d in figure 2.
- In figure 1 and page 6, I noticed that the authors believe that Cu2+ ions are embedded in the PDMS-PCDA as a 6-coordinating atmosphere. I don’t have doubt in the form of such structure, but as these Cu2+ cations are also responsible for the initiation of MOF growth (as they could serve as “fixed” metal nodes to provide anchoring point for the MOF to grow from), but since Cu2+ ions have already been saturated, how could they adopt more coordinative group such as the ones in HITP? It is for the authors good that they could prove the existence of the single PCDA coordinated Cu2+, or the actual coordinative environment of Cu2+ if all of them are actually dual-PCDA coordinated. Such information could not be provided from e.g., FTIR, unless N-Cu stretch could be proven as non-polar (symmetric). Therefore, other measurement might be needed. For example, a EPR measurement of acquired PDMS-PCDA/Cu, and PDMS-PCDA/Cu3(HITP)2 could do the work as EPR result could provide coordinative environment information to determine the spin status of Cu2+. If the result shows pure axial in PDMS-PCDA/Cu, then I totally agree that all Cu are dual-PCDA connected. But if it is axial+rhombic, then it could suggest the existence of single PCDA coordinated Cu, and further illustrating the growth mechanism. Even better, if the quantitative analysis of EPR result shows a reduced (or even changed) rhombic feature in PDMS-PCDA/Cu3(HITP)2 sample, then it directly provides the evidence of Cu3(HITP)2 growth from the single coordinated Cu2+
- In page 10 (table 1 and its discussion), as the higher rpm resulted NFA showed increased μ and ON/OFF ratio, will a even higher rpm improve them furthermore? I’m curious about the relationship between rpm & these properties, as it could be a very crucial factor for the performance improvement of such FET devices.
- Please delete redundant figure caption in line 242-245.
- Please add description of the red circle in fig.4 c, and description of fig.4 d left red box.
- Please provide full name of SAED abbreviation in line 249 (figure 4 caption). I could not find it in wither main text or SI.
- In page 10, water contact angle (CA) measurements, the CA could suggest that the surface of either PDMS-PCDA/Cu or Cu3(HITP)2 increases hydrophobicity. Conducting a CA analysis of PDMS-PCDA/Cu, PDMS-PCDA/Cu3(HITP)2, and Cu3(HITP)2 could eliminate such ambiguity. Also, a very important question is that whether the Cu3(HITP)2 MOF structure could survive, and the device still perform well after the contact of water? (means the water-stability improvement of MOF grown on the NFA, compared to the MOF grown from traditional method) Providing a positive result of such information could significantly improve the value of the authors’ approach of the fabrication of such device, and further makes this manuscript highly impactive.
- In line 318, the mechanism of O2- sounds good, but the authors do need to provide evidence to support their proposed mechanism, as there’s more than one mechanism that could explain such increase of resistance. E.g., conducting an O2-free sensing experiment to see whether resistance would increase less compared to an O2-existing environment could do the work.
- In figure 6 c, please add 2 or more data points to demonstrate the linear relationship between concentration of ethanol and response %. 4-point simulation with R2=0.98 is not very convincing. Also, if the author could provide any information of lowest LOD (limit of detection) and upper concentration boundary of linear response region, that would make the applicability of such material stronger.

Reviewer 2 Report
The authors present a strategy to grow the Cu-(HITP) MOF on oriented nanofiber arrays, which can work as electrode materials for field-effect transistor (FET) and gas sensor. The authors collect a complete analysis including XRD, TEM, SEM and other technologies, that confirms the successful growth of MOF on fibers. The device performance of FET and gas sensor has been systematically investigated. By considering the increasing interests in the development of conductive MOFs, this work is an important addition. The referee suggests its publication after addressing the following major questions.
- The author claimed that MOF@nanofiber structure was helpful for improving carrier transport. What is the proof and what is the reason behind?
- In Figure 2d (EDX mapping images of Cu), the distribution of Cu seems to be heterogeneous. How to explain the phenomena? Is this due to the formation of large MOF particles?
- The author successfully prepared the PDMS-PCDA/Cu3(HITP)2. Is it a general synthesis strategy? Can the authors vary the metal centers, for example, Ni3(HITP)2?
- In Figure 4d, the FFT did not provide any evidence about the hexagonal structure, since the TEM image only showed 100 reflection. The authors shall clarify it.
- Compared with the powder sample, the XRD pattern of Cu3(HITP)2 fiber arrays showed a stronger 001 peak while the other peaks are much weaker. Can the authors provide an detailed explanation?
- In Figure 5a, the resolution of optical microscopy image is quite low. Please provide a high-quality image.
- For FET and sensor devices, how about the performance of Cu3(HITP)2 without the nano fibers? What is the role of the nanofibers?
- Some typos should be polished. For example, in Figure 4b, the “power” should be “powder”; In line 253, the author mentioned the width is 30 nm, while in Figure 4c, the label is 50 nm.
Reviewer 3 Report
Interesting results and novelty work. A paper focuses on
In-Situ Grown Metal-Organic Framework on Oriented Nano-2 fiber Arrays for Field-Effect Transistor and Gas Sensor-Based 3 Applications, there is a few mistakes throughout the manuscript. Besides there are some grammatical mistakes throughout the manuscript, particularly in respect of use of singular and plural with the subject or verb. In view of the above comments, whole manuscript should be properly revised to make it acceptable by the journal.
- Figure 1 quality is not good it needs more high resolution one. Most of all figures and some haziness so need to improve.
2.Abstract need to involve more results not the description only
3.Title also needs to make short
4.Conclusion part be more explanatory
- The study is good and well written manuscript
